# Measuring the Correlation between Human Activity Density and Streetscape Perceptions: An Analysis Based on Baidu Street View Images in Zhengzhou, China

**Yilei Tao** [1] , **Ying Wang** [2], **Xinyu Wang** [3], **Guohang Tian** [1,4] **and Shumei Zhang** [1,4,*]

1   College of Landscape Architecture and Art, Henan Agricultural University, Zhengzhou 450002, China; yleitao@stu.henau.edu.cn (Y.T.); tgh@henau.edu.cn (G.T.)
2   College of Architecture and Engineering, Zhengzhou University of Industrial Technology, Zhengzhou 450002, China; qixian951212@163.com
3   Doctoral School of Landscape Architecture and Landscape Ecology, Technology, Hungarian University of Agriculture and Life Sciences, 1114 Budapest, Hungary; wang.xinyu@phd.uni-mate.hu
4   Henan Provincial Joint International Research Laboratory of Landscape Architecture, Henan Provincial Department of Science and Technology, Zhengzhou 450002, China
*   Correspondence: meishu_zhang83@henau.edu.cn

**Abstract:** Although investigators are using data sources to describe the visual characteristics of streets, few researchers have linked human perceptions of the street environment with human activity density. This study proposes a conceptualized analytical framework that explains the relationship between human activity density and the visual characteristics of the streetscape. The image-segmentation model DeepLabv3+ automatically extracts each pixel's semantic information and classifies visual elements from 120,012 collected panoramic street view images of Zhengzhou, China, using the entropy weighting method and weighted superposition to calculate the street perception summary score. This deep learning approach can successfully describe the semantics of streets and the connection between population density and street perception. The study provides a new quantitative method for urban planning and the development of high-density cities.

**Keywords:** human perception; human activity density; visual characteristics; urban planning; streetscape

## 1. Introduction

### 1.1. Streets as a Key Element of the Built Environment

Streets are an important part of urban public spaces [1,2]. They play a key role in the spatial organization of cities and are necessary for daily life as well [3]. Streets represent the continuation of a rich social culture. They are the showcase window of a city's image. With the rapid development of motorized transportation, the traditional street space scale and living environment have been destroyed, and more and more attention and research have been paid to urban streets. In the 1960s, western countries put forward the concept of "shared streets" [4,5], advocating the harmonious coexistence of people and vehicles and abandoning states where vehicles are the leading traffic. In 1970, the United States proposed "complete streets", and the Netherlands launched the concept of "life-oriented roads" [6,7]. What these concepts have in common is the transformation of streets from mere traffic into complex spaces of diversity. In the 1980s, the 'new urbanism' proposed that street spaces should become social places, and called for a return to the attributes of street public space by designing walkable streets and vibrant public places. The concept of street design has changed from "traffic-oriented" [8] to "human-oriented"; the function of the street has changed from single to complex; the designers of the street have changed from individual design to designs involving public participation. The public is more focused on the human experience in the street [9]. The street is the privileged place where observation takes place, from which perception is born and emotions arise (conditions which affect

human activity). However, there are differences between Chinese and western streets, which are the result of the guidance of urban planning and public policy. What is similar between them is the notion that street configuration should satisfy both functional and cultural needs. Urban cultural symbols include urban signs, urban visual guides, urban color, space environment, etc. This is a systematic process which involves refined work [10]. We are trying to find a way to meet the functions required for human activities, particularly in high-density urban environments.

### 1.2. Evolving Research Trends Promote Subject Development

As shown in Figure 1, it studied the urban anatomy that describes the city ecosystem in terms of physical structures. This classification ecosystem includes the city, its people, and the environment, which constitute an urban society [11]. People's emotions related to the social structure change with the behavior and number of people on the street [12]. For example, the street with continuous changes and the surrounding buildings and trees create a barrier giving a feeling of safety in places with dense traffic and facilities [13,14].

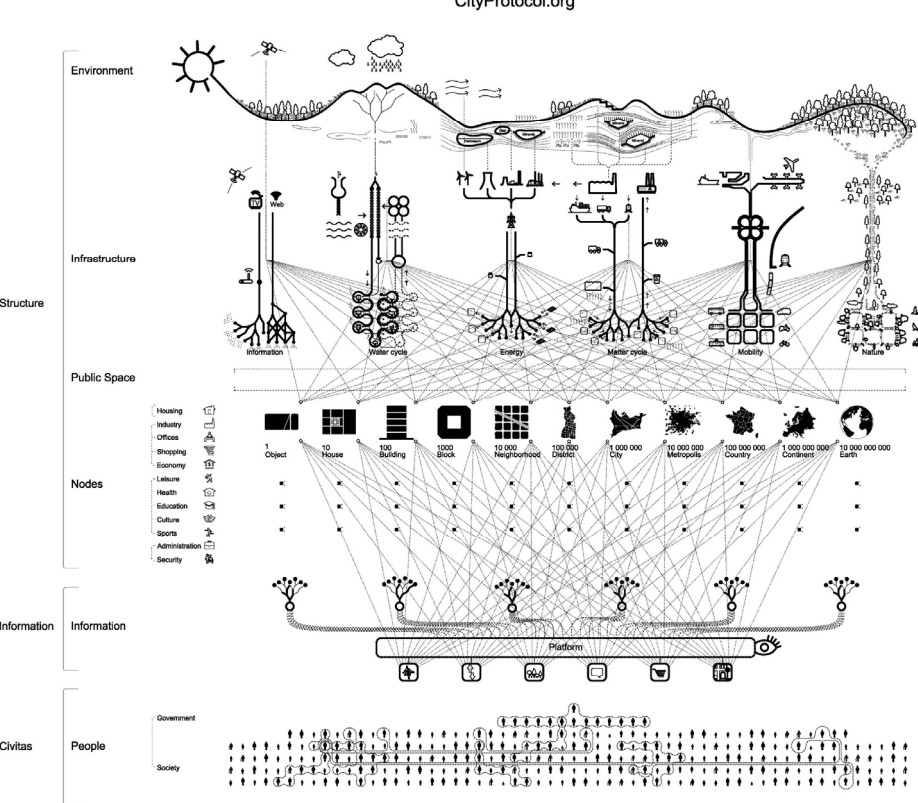

**Figure 1.** Bird's-eye view of the City Protocol Society's City Anatomy. Source: City Protocol Society.

Generally, the street environment is defined as the part of the physical space that is constructed by human activity. The street environment affects the trip mode people choose [15]. In terms of street structure, the researchers divided the street environment into traffic space, pedestrian space, green space, and enclosed the building to form a 3D space. Traffic space can be divided into vehicles and pavement; pedestrian space includes pedestrians and sidewalks; green space includes herbs, shrubs, and trees. Research on street dynamism and function, traffic characteristics, as well as human activity and perception is currently taking place in this area. Emerging big data allows these elements that define the quality of the street to be measured accurately. For instance, the quality of streets can be calculated accurately by analyzing street view images and hierarchical road networks [16]. Points of interest (POI) and smart-card records can be used to explore street functions and vitality [17,18]. Social network data and heat maps from Baidu have been used to

study human behavior [19]. Data from check-in data and mobile phone data can be used to analyze traffic characteristics [20]. Assuming that we find an adaptable range of street quality according to the distribution of human activity density, we can provide guidelines for the configuration of the streetscape in high-density cities.

### 1.3. Emerging Research Methods Help Street Environment Measurement

Humans have a multitude of senses. According to statistics, 87% of the information we obtain about the environment comes to us through vision; 11% from hearing; 3.5% from smell; 1.5% from touch; and 1% from taste. Thus, visual perception dominates the human perception of space [21]. With street view data's broad coverage and "human view" appearance, street view data have gained a great deal of attention in urban spatial research in recent years [22]. It has become the most intuitive, accurate, and effective method of observing our environment and assessing our perceptions. The most commonly used data sources are provided by map companies, including Google Street View (GSV; https://developers.google.com/maps/documentation/streetview/ (accessed on 10 November 2021)), Baidu Street View (BSV; http://lbsyun.baidu.com/index.php?title=viewstatic (accessed on 9 December 2021)) and Tencent Street View (TSV; https://lbs.qq.com/service/webService (accessed on 12 December 2021)). Researchers can obtain static street view images from these companies by using their application programming interfaces (APIs) [23].

Over the past eight years, approximately 34% of studies analyzing neighborhood environmental impressions used urban street view [24]. The walkability of streets is used to evaluate city infrastructure and amenities, and streetscape photography can quantify landscape characteristics. In addition, the connection between street vegetation, fences, and pedestrian activity differed by land type [25]. These studies have important implications for designing walkable and healthy cities [26]. Moreover, the visibility of a street has a significant impact on its ambiance. Xiaojiang Li devised a scoring system for city perception. This technique scores a large sample of semantically segmented street photos manually [27]. It is well known that streets are the most common places for people to walk, exercise, and relax daily [28]. Yi Lu states that the quality and quantity of greenery are positively correlated with the number of people in hospitals under the control of demographic factors [29]. Street view images have been used to observe the physical environment and predict a city's population and economy [30]. Alternatively, they can be used for urban planning by analyzing food, transit, retail, and housing [31]. In all studies, street view images effectively extract fundamental street components that can be utilized directly in urban physical and aesthetic studies or indirectly in urban economic and social research.

### 1.4. Quantitative Analysis Provides Scientific Standards

A street view describes an urban physical environment which is a steady state. Humans are the active elements in the street space that have variable uncertainty. In the last several years, researchers have worked to better understand these connections. The purpose of this paper is to measure the street environment, based on the perspective of human perception, in a high-density central urban area and to explore its relationship with human activity density. Therefore, the paper has two objectives: (1) to propose a multidimensional analysis framework for quantifying the percentage of each visual element in street view images (based on semantic segmentation techniques to define various aspects of scene perception); and (2) to overlay the quantified scene perception score results and human activity density data, the results of which are used to analyze spatial correlation and extremes. These methods will allow for improved urban planning by determining the peak- and low-value regions of human perception for streets with varying densities of human activity.

The remainder of this paper is organized as follows: Section 2 introduces the study area and describes the data collection. Section 3 proposes the research framework and provides the methodology. Section 4 describes the results of the study. Section 5 discusses the research results, states the findings, points out its limitations, and outlines directions

for future work. Finally, in Section 6, we sum up the conclusions of our investigation and make some constructive comments.

## 2. Study Area and Data

### 2.1. Study Area

This study site is in Zhengzhou, China (the region outlined by 112°42′ E–114°14′ E, 34°16′ N–34°58′ N), which is the capital of Henan Province and centrally located in the Central Plains City Cluster. Zhengzhou has a total area of 7511 km² and a population of 10.35 million at the end of 2019 (Henan Statistical Yearbook, 2020). To ensure the validity of the results, this study selected Zhengzhou's central districts as this study area, including the seven districts of Jinshui, Zhongyuan, Erqi, Guancheng, Huiji, Zhengdongxin, and Jingjijishukaifa (Figure 2).

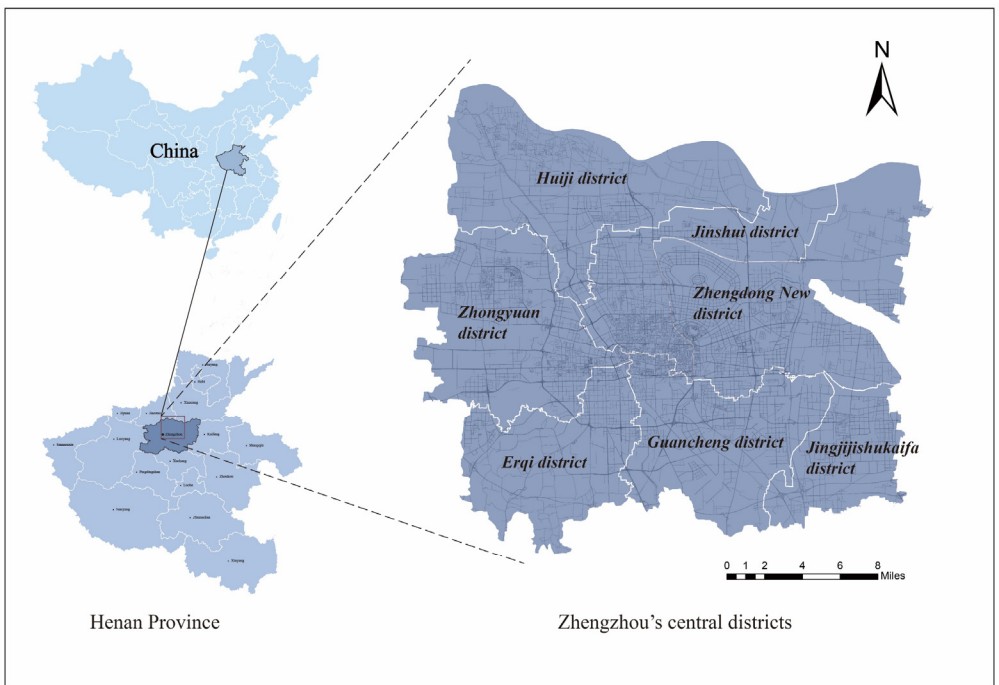

**Figure 2.** Location of the study site in Zhengzhou, China.

### 2.2. Data Collection

2.2.1. Determination of Sampling Points for the Urban Road Network

The road network of Zhengzhou was downloaded from OpenStreetMap (https://www.openstreetmap.org/#map=4/36.96/104.17 (accessed on 5 April 2020)). These road data, in vector format, were from the second quarter of 2020. The process of extracting the central road lines consisted of four steps [32,33], as follows:

1.  Initial road selection: these roads include all primary, secondary, tertiary, national, provincial, and trunk roads, as well as urban expressways, motorways;
2.  Gridding the initial roads and filling in the gaps around crossroads;
3.  Thinning the roads;
4.  Establishing sample points every 100 m along the simplified roads: finally, setting up 21,054 sample points in the study area (Figure 3a).

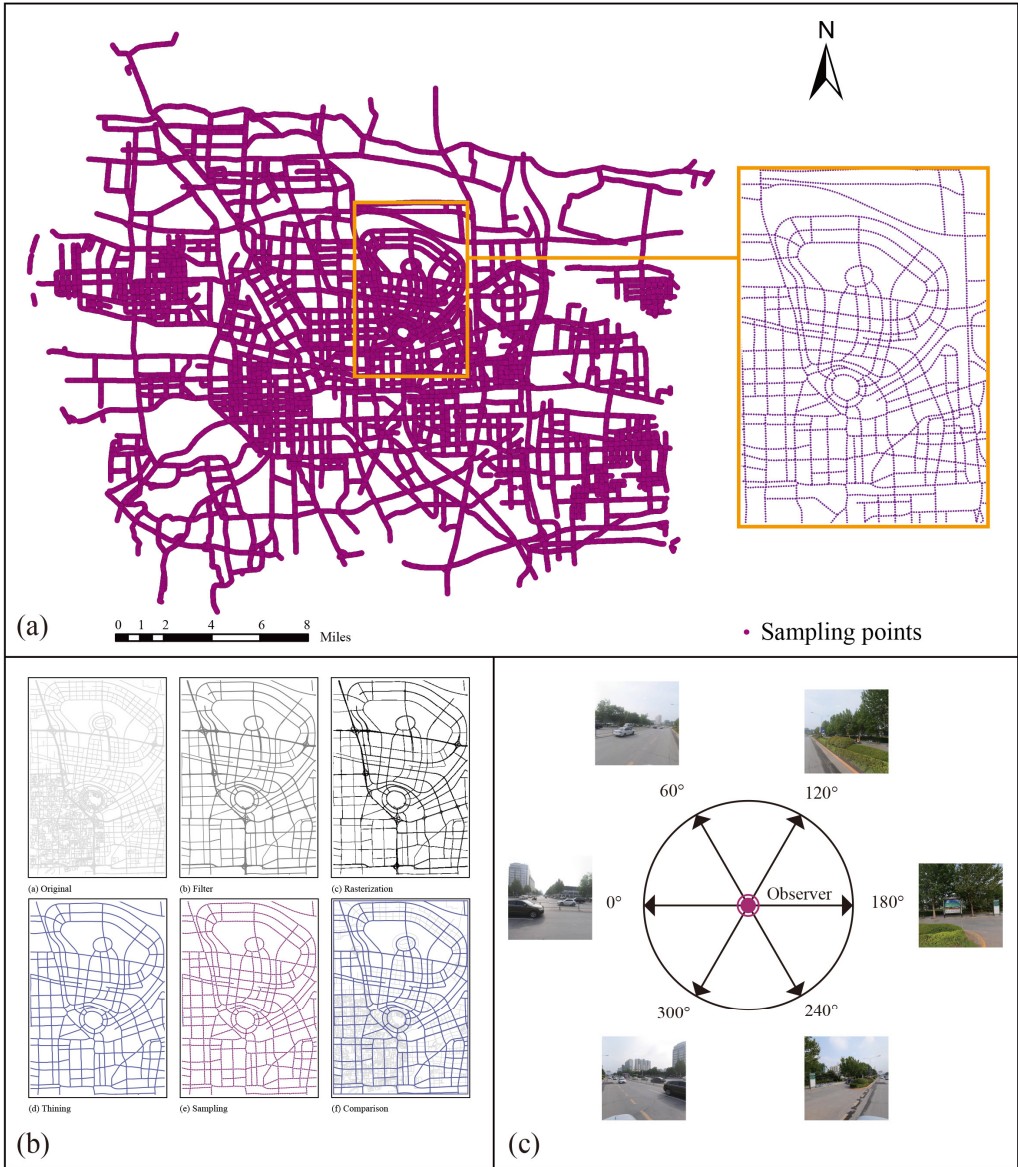

**Figure 3.** (**a**) Sampling point location and number distribution; (**b**) road network simplification procedures; (**c**) the configuration of Baidu map street view (BSV) images.

The sample points generated by these processing steps are used in the Baidu Street View (BSV) images acquisition process (Figure 3b).

### 2.2.2. Baidu Street View Image Collection

During the era of big data, Baidu Street View images represent the most significant data source to study urban streets. Several different angles of street view images are available using Baidu Maps, one of the largest online map providers in China. Since Google Maps is unavailable in China, Baidu Maps is an excellent choice with relatively high quality. This study downloaded Baidu street view (BSV) imagery using the BSV API. Each sampling point captures street view images in six radial directions at a fixed height. Using this Baidu Map data, researchers selected 20,002 sampling points, giving a total of 120,012 street view images, each with a resolution of 512 × 512 pixels, in JPG format (Figure 3c). Additionally, this study investigated the user license for Baidu location services in detail to ensure that it followed the official protocol for software development.

### 2.2.3. Conversion of Human Activity Density Data

Baidu is China's most powerful and popular search engine. Mobile phone base stations are collectively used to locate the number of users in the region, and the number of users is represented by map coloring. This information is not available to the public since it is a commercial product. However, human activity density (HAD) can be measured by observing variations in map brightness. [34,35]. During May 2017 and February 2019, four days of consecutive data were obtained via Baidu's API (Ibsyun.baidu.com) for quantitative analysis (four weekdays, four weekends). A total of 208 pieces of data were collected during the study's hour-long capture period. According to the previous article [36], the data distribution is as follows (Figure 4).

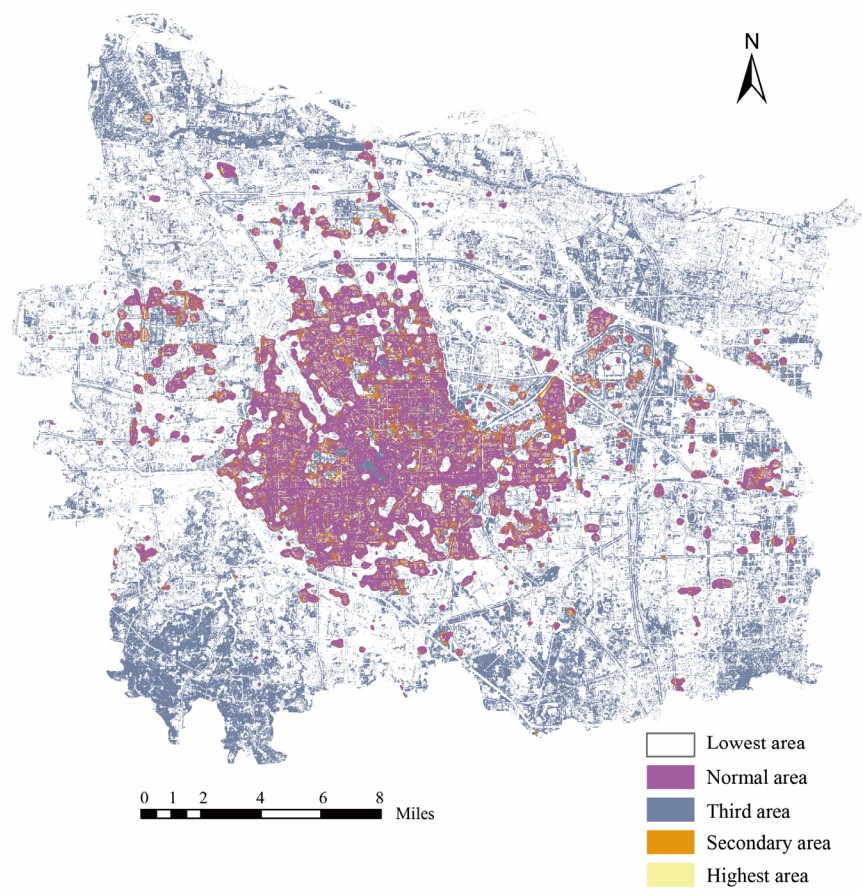

**Figure 4.** Different tiers of HAD by location (weekdays and weekends combined) for Zhengzhou, China.

## 3. Methodology

### 3.1. Conceptual Framework

This study comprises four main phases: data collection, element extraction, calculation of perception scores, and exploration of the relationship between human perception of streetscapes and HAD. Ultimately, this study can accurately locate the high- and low-value human perceptions of streetscapes with different HADs. Consequently, this study will contribute to a more efficient planning process for urban areas in the future [37]. Figure 5 illustrates the methods and analytical framework used in this paper.

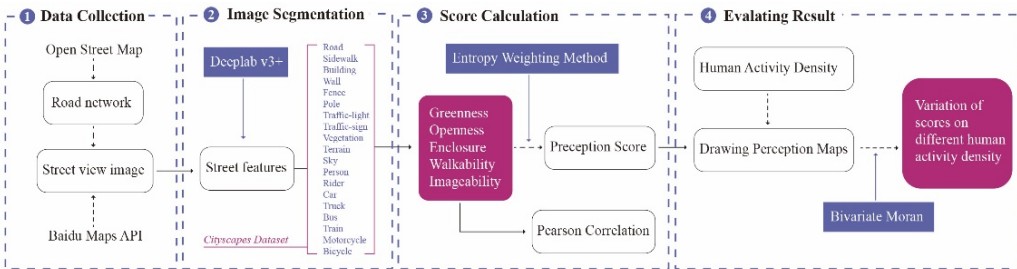

**Figure 5.** Analytical framework used in this study.

This study used five indicators to measure the human perceptions of streetscape: greenness, openness, enclosure, walkability, and imageability [38–41]. It is worth noting that the main object of what we say imageability is the street, which is guided by street traffic. Based on the combined elements of parsed visual elements from BSV images, these metrics were developed. Streetscape perception scores were represented by them. The illustrations provided in Figure 6 explain how each of the five aspects of streetscape perception works [42].

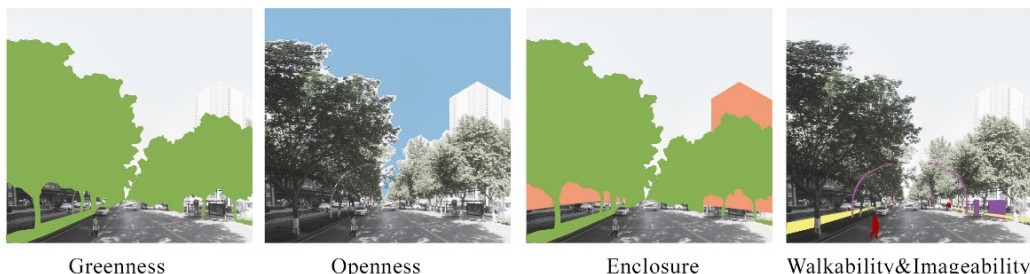

**Figure 6.** Sketches of the five aspects represent streetscape perception.

### 3.2. Classification of the Visual Elements

To investigate visual perceptions based on the BSV images, this study needs to identify the physical features of objects in each image. Therefore, researchers detected streetscape elements using the deep learning model DeepLabv3+ [43], which was developed for semantic image segmentation. The overall architecture of DeepLabv3+ is shown in Figure 7. The main body is a decoder-convolutional neural network (DCNN), with hole convolution, using common classification networks such as ResNet. A spatial pyramid pooling module called Atrous spatial pyramid pooling (ASPP) with hole convolution introduces mainly multi-scale information. The decoder module further integrates the low-level and high-level features to improve the accuracy of the segmentation boundary. When applied to cityscapes datasets, the model's effectiveness was 82.1% for the test set without post-processing [44].

This study used DeepLabv3+ trained on streetscapes from the Cityscapes dataset, which contains images with annotations of streetscape segments. The dataset is focused on the semantic representation of urban street scenes for training deep neural networks. Annotations use eight groups definition containing 30 classes based on the composition of street scenes, such as roads, sidewalks, vegetation, sky, buildings, fences, etc. The dataset provides 19 classes for training (https://www.cityscapes-dataset.com/ (accessed on 14 March 2021)), while the other 11 classes are excluded from the dataset due to rare segments in streetscapes [33].

Additionally, this study calculated the average proportion of every visual element for each sampling point, using Equation (1):

$$P_k = \frac{1}{n}\sum_{j=1}^{n} V_{jk}\{j \in (1, 2, \ldots, n), k \in (1, 2, \ldots, m)\} \tag{1}$$

where $P_k$ is the average proportion of every visual element and $V_{jk}$ denotes the proportion of visual elements in a single-direction image. $j$ denotes the number of direction images. $k$ denotes the number of the point.

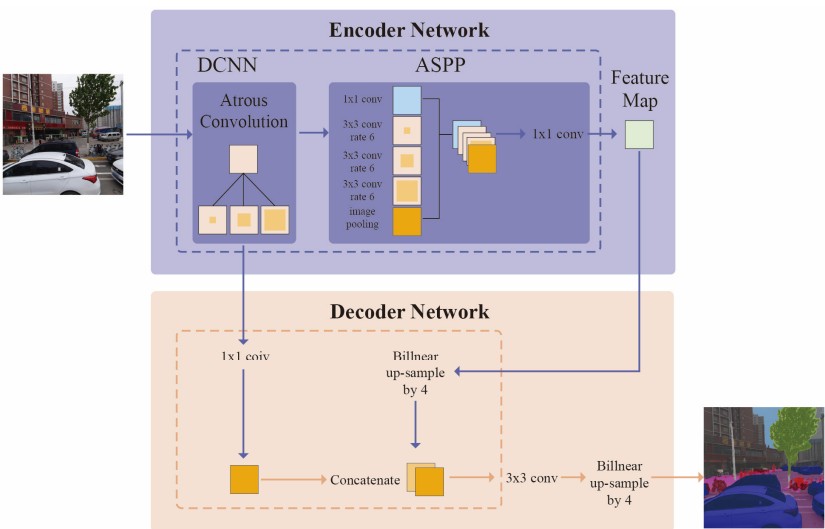

**Figure 7.** Structure of the encoder-decoder networks in the DeepLabv3+ model.

### 3.3. Calculation of Perception Scores

Table 1 in this study provides explanations of Equations (2)–(6) that were used to calculate the score of five perception indices. It can be difficult to assess precisely a visual element's permanent proportion in a scene if it includes transient and fickle elements, such as vehicles, pedestrians, and bicyclists. The calculation does not include any of these visual elements.

**Table 1.** Formulate explanations for the five indices of the perception scores.

| Indicators | Formula | | Explanation |
|---|---|---|---|
| Greenness | $G_k = \frac{1}{n}\sum_{j=1}^{n} V_{jk} + \frac{1}{n}\sum_{j=1}^{n} T_{jk}$ $\{j \in (1,2,\ldots,n), k \in (1,2,\ldots,m)\}$ | (2) | $V_{jk}$ is the percentage of pixels representing vegetation, $T_{jk}$ is the percentage of pixels representing terrain, and the sum indicates the total amount of tree pixels in each picture. |
| Openness | $O_k = \frac{1}{n}\sum_{j=1}^{n} S_{jk}$ $\{j \in (1,2,\ldots,n), k \in (1,2,\ldots,m)\}$ | (3) | $S_{jk}$ is the percentage of pixels representing sky, and the sum indicates the total amount of sky pixels in each picture. |
| Enclosure | $E_k = \frac{\frac{1}{n}\sum_{j=1}^{n} B_{jk} + \frac{1}{n}\sum_{j=1}^{n} V_{jk}}{\frac{1}{n}\sum_{j=1}^{n} R_{jk} + \frac{1}{n}\sum_{j=1}^{n} SW_{jk} + \frac{1}{n}\sum_{j=1}^{n} F_{jk}}$ $\{j \in (1,2,\ldots,n), k \in (1,2,\ldots,m)\}$ | (4) | $B_{jk}$ is the percentage of pixels representing building; $V_{jk}$ is the percentage of pixels representing vegetation, $R_{jk}$ is the percentage of pixels representing road, $SW_{jk}$ is the percentage of pixels representing sidewalk, and $F_{jk}$ is the percentage of pixels representing fence. This equation expresses how visually surrounded a street is by vertical elements and horizontal elements. |
| Walkability | $W_k = \frac{\frac{1}{n}\sum_{j=1}^{n} SW_{jk} + \frac{1}{n}\sum_{j=1}^{n} F_{jk}}{\frac{1}{n}\sum_{j=1}^{n} R_{jk}}$ $\{j \in (1,2,\ldots,n), k \in (1,2,\ldots,m)\}$ | (5) | $SW_{jk}$ is the percentage of pixels representing sidewalk, $F_{jk}$ is the percentage of pixels representing fence, and $R_{jk}$ is the percentage of pixels representing road. It indicates the visual impact of perceptible road conditions on walking experiences. |
| Imageability | $I_k = \frac{1}{n}\sum_{j=1}^{n} B_{jk} + \frac{1}{n}\sum_{j=1}^{n} TL_{jk} + \frac{1}{n}\sum_{j=1}^{n} TS_{jk}$ $\{j \in (1,2,\ldots,n), k \in (1,2,\ldots,m)\}$ | (6) | $B_{jk}$ is the percentage of pixels representing building, and $TL_{jk}$ is the percentage of pixels representing traffic-light. $TS_{jk}$ is the percentage of pixels representing traffic-sign. The equation is based on the building, sign symbol, and other street furniture that contribute to the impression of the street. |

Note: adapted from [42].

*3.4. Spatial Correlation Analysis*

3.4.1. Entropy Weighting Method

To understand more fully the human perception of streetscapes, this study carried out a superposition process. The entropy weighting method (EWM) is typically used for determining the weights of indicators [45,46]. Entropy is a measure of the degree of disorder in a system. The high entropy criteria imply a varied, unstable response with low weight.

3.4.2. Bivariate Moran's I

This study used the bivariate Moran's I to find the spatial correlation between human activity density and perception scores. As early as 2002, Anselin introduced the bivariate Moran's I, a multivariate spatial correlation indicator [47]. Later, this was implemented in popular spatial analysis tools, such as the GeoDa software [48]. This analysis mainly uses two pieces of software, ArcGIS and GeoDA. First, data processing was performed using ArcGIS to extract the values of human activity density at the sampling points. Secondly, the global Moran's I index was obtained using the Moran scatter plot in GeoDA; Finally, Local Indicator of Spatial Association (LISA) cluster plots and Moran scatter plots with the local bivariate LISA function generate fields. The LISA cluster plots show the high-high (HH), low-low (LL), high-low (HL), and low-high (LH) phenomena with significance levels.

## 4. Results

*4.1. Descriptive Statistics for the Segmented Results*

The 120,012 BSV images were processed by semantic segmentation to extract elements. Next, this study calculated the average proportion of 19 classes of visual elements using Equation (1). Table 2 summarizes the extreme, average, and standard deviation values. This study shows that the initial elements of images such as roads, sky, and vegetation have the highest proportions. They are the main components of the streetscape, impacting enclosure, walkability, openness, and greenness. The road serves as a visual element in the horizontal direction and has a load-carrying function for pedestrians, cars, etc. Its proportion is closely linked to enclosure and walkability, indicating that the study should consider installing fences to improve pedestrian safety. Since the sky is the only openness variable, its proportion shows that openness is more important in the overall assessment.

**Table 2.** Statistics for the segmented results of BSV images (%).

| Indicators | Visual Elements | Max | Min | Mean | S.D. |
|---|---|---|---|---|---|
| Initials elements of BSVIs | Road | 0.486239 | 0.000094 | 0.326304 | 0.074928 |
| | Sidewalk | 0.181760 | 0.000000 | 0.021542 | 0.025417 |
| | Building | 0.820443 | 0.000000 | 0.093750 | 0.100544 |
| | Wall | 0.409109 | 0.000000 | 0.011996 | 0.022776 |
| | Fence | 0.528089 | 0.000000 | 0.016723 | 0.024034 |
| | Pole | 0.094675 | 0.000000 | 0.005730 | 0.004817 |
| | Traffic-light | 0.010117 | 0.000000 | 0.000417 | 0.001094 |
| | Traffic-sign | 0.038918 | 0.000000 | 0.001993 | 0.002959 |
| | Vegetation | 0.810742 | 0.000000 | 0.203855 | 0.146839 |
| | Terrain | 0.336346 | 0.000000 | 0.020166 | 0.033182 |
| | Sky | 0.493867 | 0.000000 | 0.251535 | 0.127563 |
| | Person | 0.112603 | 0.000000 | 0.002230 | 0.004776 |
| | Rider | 0.038186 | 0.000000 | 0.000665 | 0.001674 |
| | Car | 0.297883 | 0.000000 | 0.028245 | 0.037712 |
| | Truck | 0.751343 | 0.000000 | 0.009446 | 0.032118 |
| | Bus | 0.312477 | 0.000000 | 0.002765 | 0.015017 |
| | Train | 0.085515 | 0.000000 | 0.000522 | 0.002597 |
| | Motorcycle | 0.068605 | 0.000000 | 0.001320 | 0.003626 |
| | Bicycle | 0.111669 | 0.000000 | 0.000829 | 0.002505 |
| Visual perception indices | Greenness | 0.843387 | 0.000000 | 0.224021 | 0.161433 |
| | Openness | 0.493867 | 0.000000 | 0.251535 | 0.127563 |
| | Enclosure | 1.000000 | 0.000097 | 0.008058 | 0.010840 |
| | Walkability | 1.000000 | 0.000000 | 0.131754 | 0.149816 |
| | Imageability | 0.821284 | 0.000015 | 0.096159 | 0.100485 |

In addition, this study randomly selected three locations with different classes. Figure 8 shows their semantic segmentation results. The images in the left column show the segmentation of 19 visual elements for semantic segmentation, and the right column shows the index of significant features. For instance, in Figure 8A, the essential visual elements are, in order, sky (43.48%), road (31.20%), and buildings (15.02%). In Figure 8C, however, the essential visual elements are, in order, sky (53.00%), vegetation (21.65%), and road (11.97%). Although the crucial components of street visual elements are sky, road, vegetation, and buildings, there are subtle differences in their proportion and importance. The road and building statistics are almost identical, indicating that the indices for walkability and imageability are stable. However, both the sky and vegetation data make up a substantial percentage of the street composition. Thus, openness, enclosure, and greenness are varied.

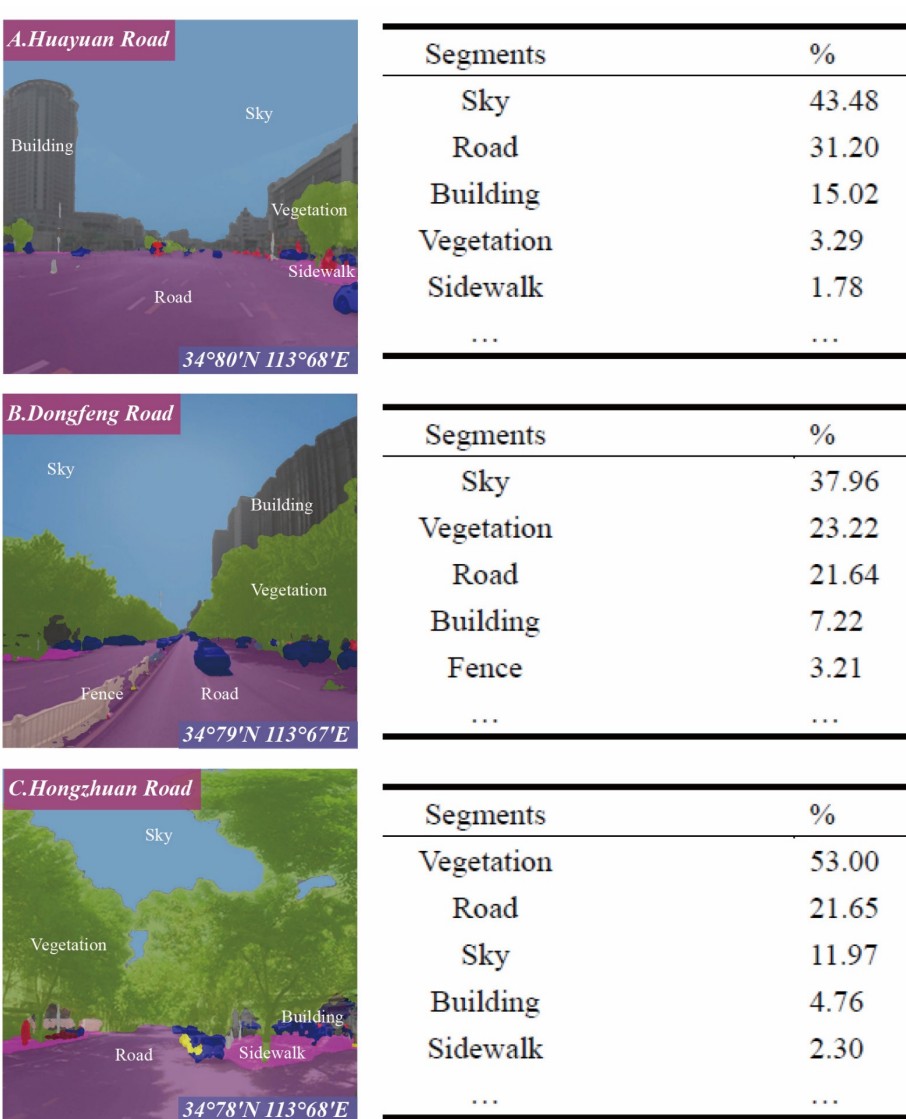

**Figure 8.** Three semantic segmentation sample cases; (**left**) images, (**right**) results.

## 4.2. Correlation Analysis of Initial Elements in BSV Images

In this study, the 20,002 individual data points collected will be tested for multicollinearity using Pearson correlation analysis. A comparison matrix of correlation coefficients is shown in Table 3 as a two-sided significance test. As demonstrated in these results, most correlation coefficients are low or negative, indicating that the visual elements used in the calculation formulas are independent and do not overlap. Multicollinearity is effectively decreased by the five perceptual indices. This indicates that the model is valid.

**Table 3.** Correlation matrices of the initial visual elements and their indices. The five indicators have been normalized.

**Part 1: Correlation coefficients between the five indices.**

| Indices | Greenness | Openness | Enclosure | Walkability | Imageability |
|---|---|---|---|---|---|
| Greenness | 1 | −0.542 ** | 0.390 ** | 0.000 | −0.358 ** |
| Openness | −0.542 ** | 1 | −0.398 ** | −0.034 ** | −0.465 ** |
| Enclosure | 0.390 ** | −0.398 ** | 1 | 0.161 ** | 0.180 ** |
| Walkability | 0.000 | −0.034 ** | 0.161 ** | 1 | 0.048 ** |
| Imageability | −0.358 ** | −0.465 ** | 0.180 ** | 0.048 ** | 1 |

**Part 2: Correlation coefficients between greenness and its visual components.**

| Indices | Greenness | Vegetation | Terrain |
|---|---|---|---|
| Greenness | 1 | 0.981 ** | 0.523 ** |
| Vegetation | 0.981 ** | 1 | 0.349 ** |
| Terrain | 0.523 ** | 0.349 ** | 1 |

**Part 3: Correlation coefficients between enclosure and its visual components.**

| Indices | Enclosure | Building | Vegetation | Road | Sidewalk | Fence |
|---|---|---|---|---|---|---|
| Enclosure | 1 | 0.179 ** | 0.382 ** | −0.467 ** | −0.009 | −0.062 ** |
| Building | 0.179 ** | 1 | −0.338 ** | −0.140 ** | 0.096 ** | 0.136 ** |
| Vegetation | 0.382 ** | −0.338 ** | 1 | −0.475 ** | 0.087 ** | −0.109 ** |
| Road | −0.467 ** | −0.140 ** | −0.475 ** | 1 | −0.236 ** | −0.152 ** |
| Sidewalk | −0.009 | 0.096 ** | 0.087 ** | −0.236 ** | 1 | 0.048 ** |
| Fence | −0.062 ** | 0.136 ** | −0.109 ** | −0.152 ** | 0.048 ** | 1 |

**Part 4: Correlation coefficients between walkability and its visual components.**

| Indices | Walkability | Sidewalk | Fence | Road |
|---|---|---|---|---|
| Walkability | 1 | 0.051 ** | 0.057 ** | −0.067 ** |
| Sidewalk | 0.051 ** | 1 | 0.048 ** | −0.236 ** |
| Fence | 0.057 ** | 0.048 ** | 1 | −0.152 ** |
| Road | −0.067 ** | −0.236 ** | −0.152 ** | 1 |

**Part 5:Correlation coefficients between imageability and its visual components.**

| Indices | Imageability | Building | Traffic-light | Traffic-sign |
|---|---|---|---|---|
| Imageability | 1 | 0.999 ** | 0.000 | 0.002 |
| Building | 0.999 ** | 1 | −0.024 ** | −0.033 ** |
| Traffic-light | 0.000 | −0.024 ** | 1 | 0.460 ** |
| Traffic-sign | 0.002 | −0.033 ** | 0.460 ** | 1 |

"**" $p < 0.010$.

Next, this study correlated the greenness, enclosure, imageability, and walkability indices with their original visual components. Since openness is calculated from a single element, its correlation is not further investigated. This study extracted the green in horizontal and vertical directions in the greenness index, vegetation, and terrain. From part two in Table 3, vegetation and terrain have a high correlation with the greening index (correlation coefficients of 0.981 and 0.523, respectively). Similarly, researchers found a strong positive correlation between building and imageability (0.999). This result suggests that buildings are an important symbol of the city and a dominant position.

In addition, this study divides the visual elements into horizontal and vertical directions from a visual perspective. In part 3 of Table 3, roads (a critical element in the horizontal direction) are negatively correlated with enclosure (−0.467). In contrast, vegetation and buildings (essential elements in the vertical direction) are positively correlated with enclosure (0.382 and 0.179, respectively), while sidewalks and fences significantly exist independently. In short, all of these factors have weak correlations with the degree

of enclosure. It was found that sidewalks, fences, and roads have a weak correlation with walkability.

To summarize, a synergistic assessment model consisting of these visual elements, such as greenness and imageability, is suitable for studying street perceptions.

### 4.3. Spatial Distribution of Human Perception Scores

To have a clearer understanding of the scores, this study carried out further spatial analysis by calculating the perception scores of the five indices for all 20,002 sampling points (Figure 9). This makes it possible in daily life and reveals some interesting findings.

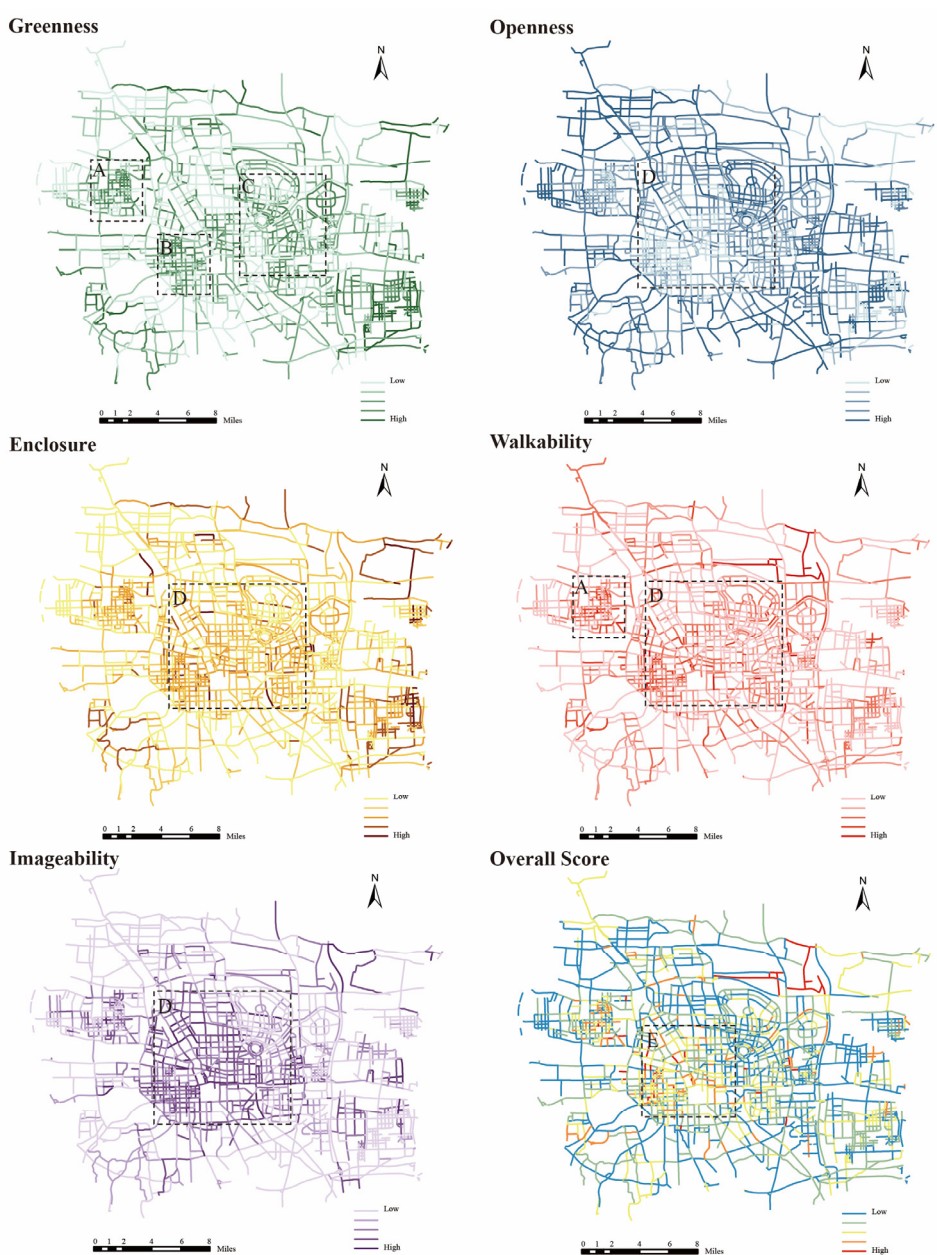

**Figure 9.** Human perception scores.

First, excluding the impact of agricultural land and woodland, three main areas of focus were determined: A, B, and C. Zone A is located in the Zhongyuan District, which was once an industrial area. For urban development, many traditional old industrial areas are being transformed into cultural and creative industrial regions. The location of Zone A contains many lush green campuses. Area B is at the junction of Zhongyuan

District and Erqi District, with high street density. The area has been maintained with high-quality greenery since the development of Zhengzhou. Area C is located in Zhengdong New District, a national comprehensive pilot zone for big data. At the same time, it has a beautiful environment with high-quality green areas. Secondly, the openness score shows an opposite distribution to the enclosure score. Overall, the central region (D) has a low openness but a high enclosure score. However, the opposite is true for the outer region. This result is related to the positioning of the area, where the streets, buildings, and population are denser in the center. Also, walkability is influenced by road and sidewalk components, with a higher score in Area D. Zones A and C also have high local scores. Finally, the imageability score map demonstrates a strong association between imageability and building density. Zone D gets a high score. And the iconic buildings in Zhengzhou, such as the Henan Art Center and the Oasis Center Building, show localized high scores.

The entropy weighting method (EWM) is applied for weighting superposition, with specific weight distributions shown in Table 4. Walkability has the highest weight factor (0.4541), followed by imageability (0.2291) and enclosure (0.1390). The entropy value represents information confusion, and the valid data better represents city streets.

**Table 4.** Entropy method weighting results.

| Indices | Entropy Value | Information Utility Value | Weighting Factor |
| --- | --- | --- | --- |
| Greenness | 0.9764 | 0.0236 | 11.11% |
| Openness | 0.9858 | 0.0142 | 6.68% |
| Enclosure | 0.9705 | 0.0295 | 13.90% |
| Walkability | 0.9036 | 0.0964 | 45.41% |
| Imageability | 0.9514 | 0.0486 | 22.91% |

As initially stated, walkability should be the most important factor from a human-centered perspective. Both imageability and enclosure are related with three or more elements, and the amount of information presented is large and has a high-weighted percentage. In contrast, openness is influenced only by the sky; openness data are variable, structurally unstable, and have the lowest weight share. The entropy weighting method avoids the interference of human factors in the assignment process. Entropy values above 95% imply that the selected indicators contain the vast bulk of the original information. As a result, the weighted superposition score for the overall street perception is reasonable. As shown in Figure 9, the overall score of human perception is represented by blue (low) to red (high). The central region E is in a prime position due to its economic, cultural, and transportation development level, so people have higher perception scores for E than other regions. This indicates that regional and neighborhood-scale city streets are more likely to receive higher scores.

In the analysis, the human perception scores of streets were classified into five levels: bad, poor, fair, good, and excellent. The percentage of streets in each category is 39.21%, 34.16%, 18.35%, 6.68%, and 1.60%, respectively. This study compared the assessment results with the Zhengzhou City Street Design Guidelines, and they are well-matched. In general, these results match classical urban design theories. High scores are often characterized by a pedestrian-friendly streetscape, such as decent building proportions, good greenery, and adequate space for pedestrians. The streetscapes that score poorly lack these key elements or do not combine them properly.

*4.4. Results of Spatial Correlation Analysis*

This analysis is divided into two parts: global and local bivariate Moran's I index calculations. The global index tells us whether there is spatial agglomeration or outliers, and the local index presents the state of the spatial distribution.

The results of the global bivariate calculation for the study area are shown in Table 5. Moran's I = 0.0347 indicates that people's perception of reception positively influences the strength of human activity density. Importantly, it needs to interpret the Moran's I based on the $p$-value ($p$) and Z-value (Z): $p$ = 0.0001, which is less than 0.05, and therefore pass

the 95% confidence test; Z = 11.7051, which exceeds the critical value of 1.65 (the threshold set by rejecting the null hypothesis).

**Table 5.** The global bivariate Moran's I index.

| I | *p* | Z | S.D. |
|---|---|---|---|
| 0.0347 | 0.0001 | 11.7051 | 0.0030 |

Figure 10 shows all of the local results. Specifically, Figure 10a shows the bivariate LISA clustering map, and Figure 10b shows the LISA significance map. LISA clustering analysis can reveal regions and anomalies due to spatial lag in the study area. In general, except for "NS," which is not significant, there are four types of clusters: "HH," "LL," "LH," and "HL." "HH" means that the variable x (Human perception score) has a high value at f (i,j), and the variable y (HAD) has a high value. In terms of spatial distribution, it can be summarized that "HH" is mainly clustered in the city center, while "LL" is the opposite; "HL" is decentralized throughout the city and accounts for less of the area. "LH" is mostly scattered all over the city but dominates the southwest corner of Zhengdong New District and the Beijing-Hong Kong-Macao Expressway (Figure 11).

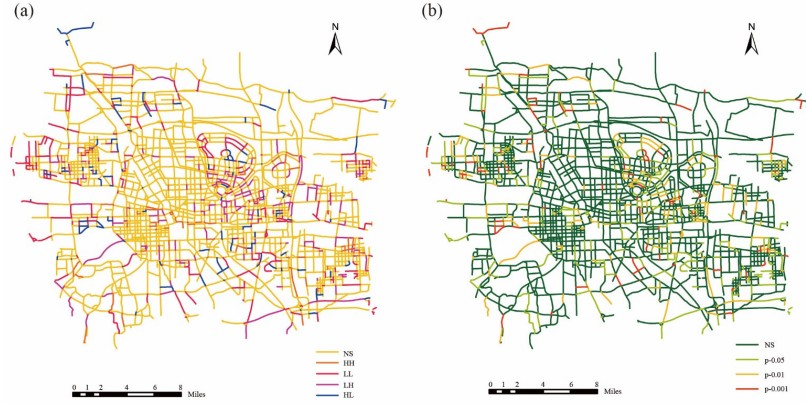

**Figure 10.** (**a**) LISA clustering map, and (**b**) LISA significance map.

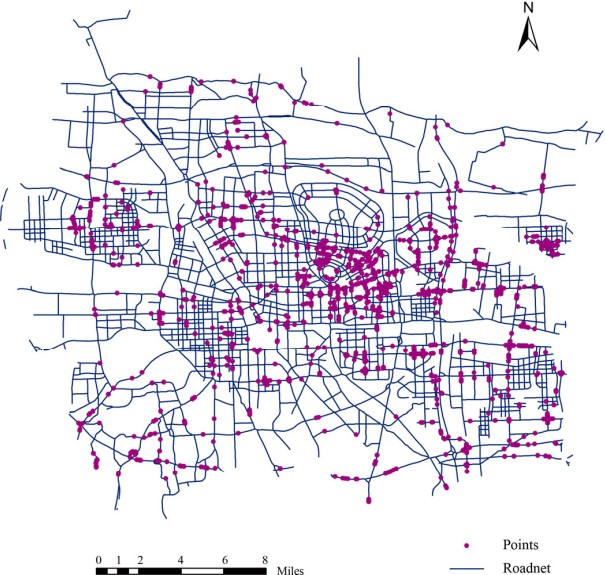

**Figure 11.** Map highlighting "LH" points in Zhengzhou, China.

This research is focused on the "LH" clusters: areas with poor human perception but excellent crowd activity. Using the analysis, one can target and perform an accurate survey for a point location in this study area and offer a feasible approach to address the problem

(i.e., a solution comparable to "Chinese acupuncture" can be put into practice). The Moran's I index is greater than zero, but the correlation is weak, with 76.8% insignificant types. In what follows, this study will discuss the reasons for this.

## 5. Discussion

Previous work has demonstrated the validity of using machine learning to study urban street perception. The study used BSV images and deep learning methods to investigate the relationship between human perception of streets and human activity density. The results indicate that walkability is an essential indicator of street design. Human activity density has a weak correlation (low or negative) with street perception scores. These findings confirm that prior urban planning lacks humanism, which ignores the relationship between population density and urban growth. Urban street design should follow the human-oriented principle, resilient development, and hierarchical shaping. The grading system is based on the level of human activity density (from high to low). The design is evaluated based on the weight of the streetscape perceptions (from high to low). With the development strategies of "slow walking, public transit, and green priority" in China, walkability will become an integral component of the future urban road transformation.

This study also analyzes the effect of the different modes of perceptions' levels on different densities. Researchers classified HAD into five levels: lowest, normal, third, secondary, and highest (Figure 4). Similarly, the five human perception indicators are classified into five levels from low to high (Figure 9). Researchers made heat maps to explore the trend of perceptions under different population intensity scenarios (Figures 12 and 13). The findings from trend research demonstrate important information for the current. When HAD is ranked lowest, the number of indicators diminishes with increasing rank, except for openness, which grows from O1 to O5. In general, the lower grades of perception are dominant. The same results were found in HAD for the normal and third grades, except that the normal grade's quantity was average. These findings are in line with the view that infrastructure is inadequate in low population density areas. The normal rank HAD has a better greenness score. However, a lack of infrastructure is certainly one of the major factors holding back development in some areas. However, when HAD was the highest and secondary rank, there was no significant tendency. Researchers observed that the number of intermediate levels perception scores is dominant (G2, G3, O3, E2, I2, I3), and walkability remains the lowest score in every grade (Figure 13). Differences in perception scores between the lowest and highest grades are a vivid demonstration of what imbalances during resource allocation.

This study gives template designs for standardized grade classification for urban planning in Zhengzhou, including precise point positioning and hierarchical shaping. Additionally, the findings from this study make several contributions to the current literature. First, the method proposed is flexible, effective, and many steps are automated. For example, this study downloads the street view images by parsing the URL using the BSV image API. Then, the information of individual elements of the photo is automatically extracted by the DeepLabv3+ semantic segmentation model. This method can be used for any city streetscape evaluation where BSV images are available and other research fields. Second, the proposed streetscape perceptions computational approach is easy to understand and use. In the future, researchers plan to analyze the five perceptions concerning street configuration and types, land use diversities, real-time traffic, etc. Incorporating more aspects, this study can explore relationships between human perception and social activities. Moreover, street view photos are regularly updated, allowing us to track changes in city streets over time, thus providing a unique viewpoint for detection approaches. Last and most importantly, this study can discover problems in urban design through this method. According to the distribution of relatively dense population activities and the visual quality of the streets, this study can propose strategies to improve and enhance the quality of space; thus, it can effectively improve human happiness. For urban planners, this study possesses essential reference value.

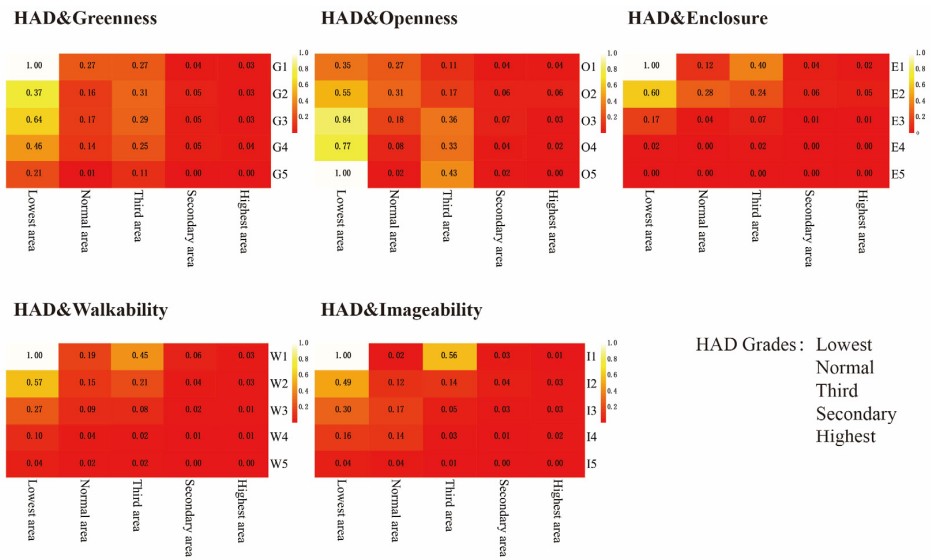

**Figure 12.** Heat map of human activity density and the five perceptions.

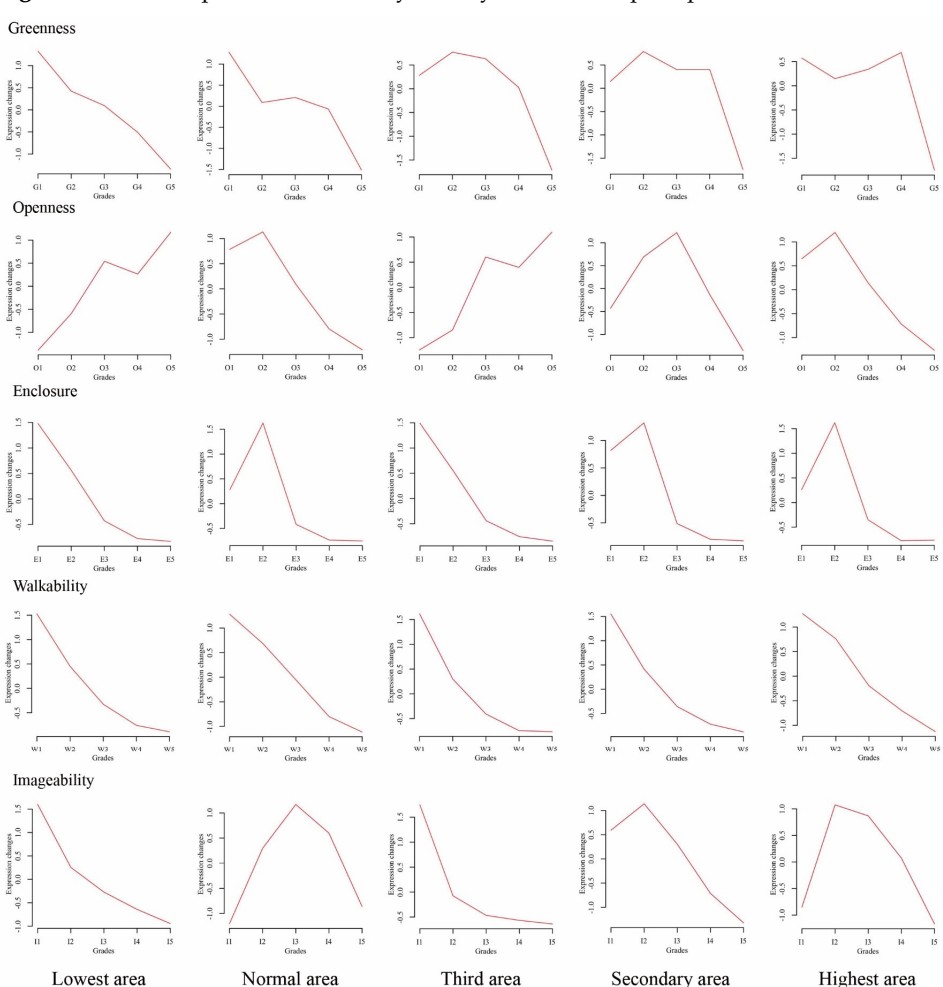

**Figure 13.** Cluster trends of different human activity densities.

However, there are some shortcomings in this study. Its major limitation is the acquisition constraints of BSV images. The onboard camera can only acquire photographs on main roads. Some small or remote streets have no street view images. So, there are some shortcoming concerning methods of manual street image collection. Fortunately, manual image collection accounts for less than 5% of the data used in this study. Also, this study

was taken within only one season. To increase generalizability, this study also needs to obtain and experiment with street scenes from other seasons. Finally, the study did not define detailed perceptions for volunteers when manually verifying the perception results. In follow-up research, it will analyze differences in the perceptions of different groups, such as different occupations and ages.

## 6. Conclusions and Policy Recommendations

This study investigated human activity density using mobile phone data and collected street view images to obtain streetscape perceptions. From these findings, this study can conclude the following:

For high-density cities, the walkability of streets is crucial. The imageability, enclosure, and greenness affected by buildings and trees are second only to walkability.

There was a weak association between HAD and street perception.

In locations with low population activity density levels, low grades dominate perceptual ratings. In locations with high levels of population activity density, perception ratings are concentrated at the medium level.

In terms of methodology, the visual dimension highlights the character of streets. It also enhances the ability of physical elements to convey visual characteristics.

Poor environmental quality is an inescapable presence in the city. Due to objective factors in society, people do not reduce activities in a place of poor quality. Therefore, researchers provide a scientific approach for urban planners and designers to create more comfortable streets and healthier cities.

Better street planning policies can be formulated by considering factors that influence human activity density when designing urban streetscapes. Given that the local government of Zhengzhou is currently in the planning stage, the study can provide some guidance for the planning of Zhengzhou streets. It recommends marking the highest density of population activity as the prioritizing transformation areas. Based on the perceived grade of the streets, low-grade areas should be fully updated and high-grade areas should be appropriately enhanced with urban furniture to improve street quality. Researchers should also consider creating more multiple public spaces by improving the properties of land use. In future research, more indicators need to be applied in areas with different population activities. This will ensure accurate assessment, including safe traffic, resilient development, and adaptability to changing local conditions.

**Author Contributions:** Conceptualization, S.Z.; methodology, Y.T.; software, Y.T.; validation, S.Z.; formal analysis, Y.T.; investigation, Y.W.; resources, G.T.; data curation, Y.T. and X.W.; writing—original draft preparation, Y.T.; writing—review and editing, S.Z.; visualization, Y.T.; supervision, G.T. and S.Z.; project administration, S.Z.; funding acquisition, S.Z. All authors have read and agreed to the published version of the manuscript.

**Funding:** This study was funded by the cultural ecosystem services in the campuses of Henan Agricultural University (30501193), the Humanities and Social Sciences Research Projects Foundation of Henan Educational Committee (2019-ZZJH-422), and the Programme of Introducing Talents of Discipline to Universities (CXJD2021004).

**Data Availability Statement:** The data presented in this study are available on request from the first author.

**Conflicts of Interest:** No potential conflict of interest were reported by researchers.

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
