# Peer review of "Measuring the Correlation between Human Activity Density and Streetscape Perceptions: An Analysis Based on Baidu Street View Images in Zhengzhou, China"

_land, doi:10.3390/land11030400_

Round 1
Reviewer 1 Report
This article presents a solid study both on the proposed conceptual framework and on the results it allows to establish. It would certainly be interesting to use the same framework and apply it to other urban areas, elsewhere in China and also in other countries.
Author Response
Thank you for your comments concerning our manuscript. Your affirmation is the greatest encouragement to our research. We tried our best to improve the manuscript and made some changes in the revised paper which will not influence the content and framework of the paper. We appreciate you.

Reviewer 2 Report
The article presents an interesting approach in using of data sources to describe the visual characteristics of streets. In the original aims of an analytical framework between human activity density and the visual characteristics of the streetscape, the individuation of a street semantics proposes the results as an interesting quantitative method for the urban planning and development of high-density cities, integrable to GIS approach.
Anyway, if the digital processes presents an operative results, it is important to review the introduction and some elements in the paper.
About introduction, in particular the section 1.1, the start of the paper is paradigmatic: "City streets are a focal point of human activity in city centers": it is almost pleonastic ... the road since ancient times before the Roman Empire is the place of human activities, is the very genesis of the city that arises from the street (and not vice versa) .Even if we understand that the introduction try to interpret the role of the street, it must be emphasized that the development of this scope runs into trivial approximations and misleading, such as the reference to orthogonality, a limiting condition that does not represent a necessary hypothesis for the proposed path, or as in 1.2, the conceptualization of the historical evolution of the streets in the city, which presents substantial trivializations of a much more complex process marked for example by an unscheduled increase in the volume of vehicular traffic over the years.
Firstly I suggest to define more clearly that the object of study are the streets within the city, marking an issue reported in the bibliography that has been interested in it. In this regard, among other things, beyond historical reflections, it is suggested to underline the car revolution and the value of Post Modern "on the road" culture and the teachings of Robert Venturi made hyperbolic by the case of Las Vegas. Again to define the theme, the difference in the perception of the environment by walk should be highlighted. In the first part, the street in the city must emerge as the defined theme, the privileged place where observation takes place, from which perception is born and emotions arise, a condition that affects human activity.
Subsequently, the second part of the introduction should be to define the hypotheses underlying the thesis, then identify what are the conditions that lead to the validity of the approach used.
I therefore want to suggest with great proactivity to place the emphasis as first point on the value of the image, the theme of the landscape, the perception of the city, which occurs mainly from the street and mainly through slow paths, “walkability, human emotion perception, noise, perception, sense of vitality, and perception of health”, all key themes central to the paper mentioned but which must be explored with the corresponding literature. In this context it is necessary to deepen the theme of visuality, responsible for about 80% of our sensations: by taking up Gregory's studies between the eye and the brain, it is possible to focus on the image the connecting element between perception and the proposed digital processes.
The second point may be the theme of the "complexity and contradiction" of the city, of the landscape, from which the hypothesis of a deconstruction by elements derives. In this sense, as Figure 1 clearly expresses, it is correct to highlight the logic of selecting characteristic elements. In this regard, I suggest to report those urban models linked in particular to the territorial and historical situations mentioned, which perhaps need to be analyzed more schematically if considered fundamental for the analysis. It could be suggested to move here “Moreover, the visibility of a street has a significant impact [….] Alternatively, they can be used for urban planning by analyzing food, transit, retail, an…”. However, it is important that the interpretative premise is operational, and for example the proposed subdivision of the roads themselves, certainly congruous, could be projected to a different evaluation of the data.
The third is the validity and transferability of the results developed in China, highlighting the points of contact but also the differences between the roads in the West and the East, in particular for that culture of signs over architecture resulting from processes of globalization but also of previous local culture.
The fourth point is the scope of the process, its democratization, the use of images in open source, the alternative or supplementary role to GIS, marked by interpretations on the vertical plane, correlated to our vision, therefore suitable for representing perception, where the residual point 1.3 is inserted.
Point 1.4 thus becomes a final explanation of the path, correctly defined.
If on section 2 there are no substantial requests, for the methodology it can be emphasized that the index used proposed is a criterion, which must not be demonstrated but only verified in its impact. It is advisable to reflect on this condition.
We also want to point out that the term “Immaginabilaty” used for buildings, traffic lights and signals is completely misleading, referring to the literature of Kevin Lynch (not quoted). It is advisable to take another term, probably from the studies on Learning from Las Vegas.
It is recommended to not use hyphenation.
Author Response
Response to Reviewer 2 Comments
Point 1: “About introduction,… or as in 1.2, the conceptualization of the historical evolution of the streets in the city, which presents substantial trivializations of a much more complex process marked for example by an unscheduled increase in the volume of vehicular traffic over the years.”
“Firstly I suggest to define more clearly that the object of study are the streets within the city,… has been interested in it. “
“In this regard, among other things, beyond historical reflections, it is suggested to underline the car revolution and the value of Post Modern "on the road" culture…”
“Again to define the theme, the difference in the perception of the environment by walk should be highlighted...human activity.”
Response 1: We have changed the introduction of the article according to your suggestion.
1.1, first, we propose that the streets within the city are important places for human life, highlighting the theme.
Please see the page 1 line 28 to 30, as following:
” Streets as an important part of urban public space [1, 2]. They play a key role in the spatial organization of cities and are necessary for daily life as well [3]. Streets represent the continuation of a rich social culture. They are the showcase window of the city image.”
Second, we presented the history of the development of the street from 1960. In high-density areas, we found rail transit and walking were the main modes of transportation due to an increase in vehicular traffic.
Please see the page 1 line 31 to 44, as following:
” With the rapid development of motorized transportation, the traditional street space scale and living environment have been destroyed, and more and more attention and research have been paid to urban streets. In the 1960s, western countries put forward the concept of "shared streets" [4, 5], advocating the harmonious coexistence of people and vehicles, and abandoning the state where vehicles are the leading traffic. In 1970, the United States proposed "complete streets", and the Netherlands launched the concept of "life-oriented roads" [6, 7]. What these concepts have in common is the transformation of streets from mere traffic into complex spaces of diversity. In the 1980s, the new urbanism proposed that street space should become a social place, and called for a return to the attributes of street public space by designing walkable streets and vibrant public places. The concept of street design has changed from "traffic-oriented" [8] to "human-oriented"; the function of the street has changed from single to complex; the designers of the street have changed from individual design to public participation. The public is more focused on the human experience in the street [9].”
Third, again to define the theme.
Please see the page 2 line 44 to 45, as following:
” The street is the privileged place where observation takes place, from which perception is born and emotions arise a condition that affects human activity.”
Point 2: “Subsequently, the second part of the introduction should be to define the hypotheses underlying the thesis, then identify what are the conditions that lead to the validity of the approach used.”
Response 1: Yes, I agree with your comments. As you suggested, we have made hypotheses in 1.2. Please see page 2 line 72 to 74, as following:
“Assuming that we find the adaptable range of street quality according to the distribution of human activity density, we can provide guidelines for the configuration of the streetscape in high-density cities.”
Point 3: “I therefore want to suggest with great proactivity to place the emphasis as first point on the value of the image, the theme of the landscape, the perception of the city, …by taking up Gregory's studies between the eye and the brain, it is possible to focus on the image the connecting element between perception and the proposed digital processes.”
“The second point may be the theme of the "complexity and contradiction" of the city, of the landscape, from which the hypothesis of a deconstruction by elements derives. …and for example the proposed subdivision of the roads themselves, certainly congruous, could be projected to a different evaluation of the data.”
Response 3: According to your suggestion, we have revised the introduction.
1.1, streets as a key element of the built environment.
1.2, evolving research trends promote subject development.
First, we start with City Protocol Society to explore the spatial background of the street. we move the Figure 1 from 1.1 to 1.2. And we recognize the complexities and contradictions of the city.
Please see page 2 line 53 to 66, as following:
“As shown in Figure 1, it studied the urban anatomy that describes the city ecosystem in terms of physical structures. This classification ecosystem includes the city, its people, and the environment, which constitute an urban society [11]. People’s emotions related to the social structure change with the behavior and number of people on the street [12]. For example, the street with continuous changes and the surrounding buildings and trees create a barrier giving a feeling of safety in places with dense traffic and facilities [13, 14].
Generally, the street environment is defined as the part of the physical space that is constructed by human activity. The street environment affects the trip mode people choose [15]. In terms of street structure, the researchers divided the street environment into traffic space, pedestrian space, green space, and enclosed the building to form a 3D space. Traffic space can be divided into vehicles and pavement; pedestrian space includes pedestrians and sidewalks; green space includes herbs, shrubs, and trees. Research on street dynamism and function, traffic characteristics, as well as human activity and perception is taking place in this area.”
Second, by learning POI, social network data, heat maps from Baidu, street view image…can help us measure these properties of streets.
Please see page 2 line 66 to 71, as following:
”Emerging big data allows these elements that define the quality of the street to be measured accurately. For instance, the quality of streets can calculate accurately by analyzing street view images and hierarchical road networks [16]. Points of interest (POI) and smart-card records can be used to explore street functions and vitality [17, 18]. Use social network data, heat maps from Baidu, to study human behavior [19]. Data from check-in data and mobile phone data can be used to analyze traffic characteristics [20].”
1.3, emerging research methods help street environment measurement.
First, we emphasize the importance of vision in the senses.
Please see page 3 line 78 to 81, as following:
“Humans have a multitude of senses. According to the statistics, the senses we obtain about the environment are 87% from vision, 11% from hearing, 3.5% from smell, 1.5% from touch, and 1% from taste. Thus, visual perception dominates the human perception of space [21].”
Second, we propose the value of street view images and the introduction.
Please see page 3 line 81 to 89, as following:
“With street view data's broad coverage and "human view" appearance, street view data has gained a great deal of attention in urban spatial research in recent years [22]. It becomes the most intuitive, accurate, and effective method of observing our environment and assessing our perception. The most commonly used data sources are provided by map companies, including Google Street View (GSV; https://developers.google.com/maps/documentation/streetview/ ), Baidu Street View (BSV; http://lbsyun.baidu.com/index.php?title=viewstatic ) and Tencent Street View (TSV; https://lbs.qq.com/service/webService ). Researchers can obtain static street view images from these companies by using their Application Programming Interfaces (APIs)[23].”
Last, the focus of the article is on the application of street view images. We illustrate the feasibility of the method with many studies.
Please see page 3 line 90 to 106, as following:
“Over the past eight years, approximately 34% of studies analyzing neighborhood environmental impressions used urban street view[24]. The walkability of streets is used to evaluate city infrastructure and amenities, and streetscape photography can quantify landscape characteristics. In addition, the connection between street vegetation, fences, and pedestrian activity differed by land type[25]. These studies have important implications for designing walkable and healthy cities[26]. Moreover, the visibility of a street has a significant impact on its ambiance. Xiaojiang Li devised a scoring system for city perception; this technique scores a large sample of semantically segmented street photos manu-ally[27]. It is well known that streets are the most common place for people to walk, exercise, and relax daily[28]. Yi Lu states that the quality and quantity of greenery are positively correlated with the number of people in hospitals under the control of demographic factors[29]. Street view images have been used to observe the physical environment and predict the city’s population and economy[30, 31]. Alternatively, they can be used for urban planning by analyzing food, transit, retail, and housing [32]. In all studies, street view images effectively extract fundamental street components that can be utilized directly in urban physical and aesthetic studies or indirectly in urban economic and social research.”
1.4, quantitative analysis provides scientific standards. It becomes a final explanation of the path, correctly defined.
Point 4: ” The third is the validity and transferability of the results developed in China, highlighting the points of contact but also the differences between the roads in the West and the East, in particular for that culture of signs over architecture resulting from processes of globalization but also of previous local culture.”
Response 4: It was rewritten.
Please see page 2 line 46 to 51, as following:
“However, there are differences between Chinese and Western streets, which are the result of the guidance of urban planning and public policy. The same thing is that street configuration should satisfy both functional and cultural needs. Urban cultural symbols include urban signs, urban visual guides, urban color, space environment, etc. This is a systematic process and refined work.”
Point 5: “The fourth point is the scope of the process, its democratization, the use of images in open source, the alternative or supplementary role to GIS, marked by interpretations on the vertical plane, correlated to our vision, therefore suitable for representing perception, where the residual point 1.3 is inserted.”
Response 5: We have proposed the use of images in open source with many studies in 1.3.
Please see page 3 line 90 to 106, as following:
“Over the past eight years, approximately 34% of studies analyzing neighborhood environmental impressions used urban street view[24]. The walkability of streets is used to evaluate city infrastructure and amenities, and streetscape photography can quantify landscape characteristics. In addition, the connection between street vegetation, fences, and pedestrian activity differed by land type[25]. These studies have important implications for designing walkable and healthy cities[26]. Moreover, the visibility of a street has a significant impact on its ambiance. Xiaojiang Li devised a scoring system for city perception; this technique scores a large sample of semantically segmented street photos manu-ally[27]. It is well known that streets are the most common place for people to walk, exercise, and relax daily[28]. Yi Lu states that the quality and quantity of greenery are positively correlated with the number of people in hospitals under the control of demographic factors[29]. Street view images have been used to observe the physical environment and predict the city’s population and economy[30, 31]. Alternatively, they can be used for urban planning by analyzing food, transit, retail, and housing [32]. In all studies, street view images effectively extract fundamental street components that can be utilized directly in urban physical and aesthetic studies or indirectly in urban economic and social research.”
Point 6: “If on section 2 there are no substantial requests, for the methodology it can be emphasized that the index used proposed is a criterion, which must not be demonstrated but only verified in its impact. It is advisable to reflect on this condition.”
Response 6: The calculation of population activity density is the status of the in real-time. We use the Jenks natural breaks classification method to classify, we can tested by the location of highest HAD of the result, these places are almost consistent with the results of our field research. High means the highest value(the higher the brightness value was, the greater the activity density). This is the method that is accepted by the public. In the absence of standards, it is a more accurate way of expression. In future studies, we will try to avoid this situation as much as possible.
Point 7: We also want to point out that the term “Immaginabilaty” used for buildings, traffic lights and signals is completely misleading, referring to the literature of Kevin Lynch (not quoted). It is advisable to take another term, probably from the studies on Learning from Las Vegas.
Response 7: We have added some references to explain “Imageability”.
Please see page 6 line 188, as following:
“This study used five indicators to measure the human perceptions of streetscape: greenness, openness, enclosure, walkability, imageability [39-42]. It is worth noting that the main object of what we say imageability is the street, which is guided by street traffic.”
First, Kevin Lynch defines imageability as a quality of a physical environment that evokes a strong image in an observer. Landmarks are believed to be a key component of imageability. The term ‘landmark’ does not necessarily denote a grandiose civic structure or even a large object. Distinctive buildings are the most common type of landmarks. Memorable buildings are characterized by complex shapes, large sizes, and high use. Additional elements that may enhance building recall are natural features around them, ease of pedestrian access, and uniqueness of architectural style.
Second, the author of “LEARNING FROM LAS VEGAS” Robert Venturi believes that architecture is an important symbol. And the main object of what we call imageability is the street, which is guided by street traffic.
Third, correlation coefficients between imageability and its visual components show that building has an extremely high correlation (0.999) with imageability, while other elements are negligible.
Please see page 12 table 3 part 5, as following:
And because of some limitations, the dataset provides 19 classes for training (https://www.cityscapes-dataset.com/ ), while the other 11 classes are excluded from the dataset due to rare segments in streetscapes. We have selected the elements that represent distinctive buildings and sign symbols to illustrate imageability by calculating.
Last, many articles use a similar method. It shows that it is feasible and practical.
For example, Ma, X.; Ma, C.; Wu, C.; Xi, Y.; Yang, R.; Peng, N.; Zhang, C.; Ren, F., Measuring human perceptions of streetscapes to better inform urban renewal: A perspective of scene semantic parsing. Cities 2021, 110, 103086 doi:10.1016/j.cities.2020.103086
Dai, L.; Zheng, C.; Dong, Z.; Yao, Y.; Guan, Q., Analyzing the correlation between visual space and residents' psychology in Wuhan, China using street-view images and deep-learning technique. 2021,
Point 8: It is recommended to not use hyphenation.
Response 8: We have used the manuscript template provided by the journal.
Thank you for your comments concerning our manuscript. We tried our best to improve the manuscript and made some changes in the revised paper which will not influence the content and framework of the paper. We appreciate you.

Reviewer 3 Report
The subject of the article is interesting and useful in making decisions regarding spatial policy. The Authors used five indicators to measure the human perceptions of streetscape: greenness, openness, enclosure, walkability, and imageability, which were classified into five levels from low to high. They also classified human activity density (HAD) into five levels, that is, lowest, normal, third, secondary, and highest. Then, a spatial correlation analysis was performed between these levels. The approach adopted by the authors is an interesting extension and supplementation of the methods used so far.
The research method has been clearly described. In my opinion, the approach adopted by the authors is interesting, has potential, and can also be used in other issues. The text is consistent and well structured. The introduction contains a well-described and exposed the goal of the research. The order of individual parts also does not raise any objections.
The content of the manuscript is consistent with its title. The authors also cited many sources from the literature, which I consider valuable in this scope.
However, I found minor editorial shortcomings in the text:
- In my opinion, formulas (1) to (6) (Table 1) are not correctly written. Please, explain what the indices "i" and "n" mean in these formulas. I suppose that on the right side of the formulas there should be the designations Vi, Ti, Si, Bi, ... instead of Vn, Tn, Sn, Bn, ... (you sum by index "i", not "n"). Then what does the index "i" mean on the left side of the formulas? Perhaps an additional index should be used here? I know that these formulas were exactly taken from the publication [34] (Ma, X.; Ma, C.; Wu, C.; Xi, Y.; Yang, R.; Peng, N.; Zhang, C.; Ren, F., Measuring human perceptions of streetscapes to better inform urban renewal: A perspective of scene semantic parsing. Cities 2021, 110). However, in my opinion, the imprecise notation of these formulas should be corrected.
- The quality of Figure 12 needs to be improved, as the values in the matrix cells are difficult to read.
Dear Authors, I recommend publishing the manuscript after making minor editing changes to the text.
Author Response
Response to Reviewer 3 Comments
Point 1: Formulas (1) to (6) (Table 1) are not correctly written. Please, explain what the indices "i" and "n" mean in these formulas. I suppose that on the right side of the formulas there should be the designations Vi, Ti, Si, Bi, ... instead of Vn, Tn, Sn, Bn, ... (you sum by index "i", not "n"). Then what does the index "i" mean on the left side of the formulas? Perhaps an additional index should be used here? I know that these formulas were exactly taken from the publication [34] (Ma, X.; Ma, C.; Wu, C.; Xi, Y.; Yang, R.; Peng, N.; Zhang, C.; Ren, F., Measuring human perceptions of streetscapes to better inform urban renewal: A perspective of scene semantic parsing. Cities 2021, 110). However, in my opinion, the imprecise notation of these formulas should be corrected.
Response 1: The formulas have been rewritten.
We have rewritter this part according to the Reviewer’s suggestion.
Whereis the average proportion of every visual element.
denotes the proportion of visual elements in a single-direction image.
denotes the number of direction images.denotes the number of the point.
in this text means six angles.denotes the number of the sampling point in this paper.
The above explanation also applies to Equation 2-6.
Point 2: The quality of Figure 12 needs to be improved, as the values in the matrix cells are difficult to read.
Response 2: We have changed picture 12 with a higher resolution image, and the values in the matrix cells are clear.
Thank you for your comments concerning our manuscript. Your affirmation is the greatest encouragement to our research. We tried our best to improve the manuscript and made some changes in the revised paper which will not influence the content and framework of the paper. We appreciate you.

Round 2
Reviewer 2 Report
Good work! Please just review the first sentence!
This manuscript is a resubmission of an earlier submission. The following is a list of the peer review reports and author responses from that submission.